# Solubilization of Paclitaxel with Natural Compound Rubusoside toward Improving Oral Bioavailability in a Rodent Model

**DOI:** 10.3390/pharmaceutics16081104

**Published:** 2024-08-22

**Authors:** Jian Zhang, Jicheng Shu, Rhett W. Stout, Paul S. Russo, Zhijun Liu

**Affiliations:** 1School of Renewable Natural Resources, Louisiana State University Agricultural Center, Baton Rouge, LA 70803, USA; jianzhang@sina.com (J.Z.); shujc210@jxutcm.edu.cn (J.S.); 2School of Perfume and Aroma Technology, Shanghai Institute of Technology, Shanghai 201418, China; 3Key Laboratory of Modern Preparation of TCM, Jiangxi University of Traditional Chinese Medicine, Ministry of Education, Nanchang 330004, China; 4Department of Pathological Sciences, School of Veterinary Medicine, Louisiana State University, Baton Rouge, LA 70803, USA; rstout1@lsu.edu; 5Department of Materials Science, Department of Chemistry and Biochemistry, Georgia Institute of Technology, Atlanta, GA 30332, USA; paul.russo@mse.gatech.edu

**Keywords:** cannulation, nano-micelles, permeability, pharmacokinetics, Taxol^®^, time–concentration curve

## Abstract

Paclitaxel, which features low water solubility and permeability, is an efflux pump substrate. The current paclitaxel drugs are given intravenously after resolving the solubility issue. Yet, oral delivery to achieve therapeutic bioavailability is not effective due to low absorption. This study evaluated a natural compound, rubusoside, to improve oral bioavailability in an animal model. Free paclitaxel molecules were processed into nano-micelles formed in water with rubusoside. The particle size of the nano-micelles in water was determined using dynamic light scattering. The oral bioavailability of paclitaxel in nano-micelles was determined against Cremophor/alcohol-solubilized Taxol after oral and intravenous administration to pre-cannulated Sprague Dawley rats. When loaded into the rubusoside-formed nano-micelles, paclitaxel reached a supersaturated concentration of 6 mg/mL, 60,000-fold over its intrinsic saturation of 0.1 µg/mL. The mean particle size was 4.7 ± 0.7 nm in diameter. Compared with Taxol^®^, maximum blood concentration was increased by 1.5-fold; the time to reach maximum concentration shortened to 0.8 h from 1.7 h; and, relative oral bioavailability increased by 88%. Absolute oral bioavailability was 1.7% and 1.3% for the paclitaxel nano-micelles and Taxol^®^, respectively. Solubilizing paclitaxel with rubusoside was successful, but oral bioavailability remained low. Further inhibition of the efflux pump and/or first metabolism may allow more oral paclitaxel to enter systemic circulation.

## 1. Introduction

Paclitaxel, a diterpenoid pseudoalkaloid and an anti-tumor agent that was first isolated from the bark of the Pacific yew tree (*Taxus brevifolia*) in the early 1970s is a chemotherapy medication to treat several types of solid tumor cancers, including ovarian, breast, and lung [1]. Sold under the brand name Taxol^®^, Paxene™, and Abraxane^®^, paclitaxel is the most effective chemotherapeutic natural compound with inhibitory concentrations in the lower nanomolar range. It is included in the World Health Organization’s List of Essential Medicines. All current medications of paclitaxel are administered intravenously. An oral medication would be highly desirable.

The lack of oral paclitaxel formulation may be explained by its physicochemical properties and biological behavior in the human gastrointestinal tract (GIT) system. In the biopharmaceutical classification system (BCS), paclitaxel is classified as a BSC IV drug, the most difficult category for oral drug development [2]. BSC IV drugs feature poor solubility and low permeability. Poor solubility limits access to the absorption villi in the small intestine, whereas low permeability affects intestinal absorption [3]. Low permeability is due to paclitaxel serving as a substrate of the P-glycoprotein (P-gp) enzyme [4,5,6,7,8,9]. The poor solubility and low permeability have seriously limited the clinical application of paclitaxel [10].

The use of rubusoside, a natural diterpene glycoside isolated from the leaves of *Rubus suavissimus*, native to the Guangxi region of southwestern China, has seemingly solved both the poor solubility and low permeability problems. In our previous study, rubusoside solubilized paclitaxel to a concentration of 6.3 mg/mL [3], comparable to the Taxol^®^ drug concentrate of 6 mg/mL. Unlike the Taxol^®^ drug, rubusoside-solubilized paclitaxel (Nano-PTX) was in the form of water-soluble powder. When reconstituted in water, a water-soluble concentrate was formed, which was dilutable from 6258 µg/mL to as low as 76 µg/mL. In a stability evaluation, the reconstituted concentrate and its serial dilutions remained soluble and stable for at least 24 h in saline as well as physiologic stomach and intestinal solutions. The water-soluble paclitaxel powder had a simple composition: rubusoside and paclitaxel, only two ingredients and free of organic solvent, emulsifier, and other formulating ingredients. In a side-by-side comparison in a Caco-2 cell culture, paclitaxel at a concentration of 60 µg/mL was 3.5 times more permeable in the Nano-PTX formulation than in the Taxol^®^ formulation. In addition, the safety profile of rubusoside is excellent because it is generally regarded as safe (GRAS) for food sweetening. These results and prospects raised the hope of reformulating paclitaxel with a natural GRAS compound to deliver it orally.

The Caco-2 results only serve to predict oral absorption since they answer a simple permeability question. This present study follows up on that promise with a validating animal study; specifically, it compares the Nano-PTX formulation with the Taxol^®^ formulation in a Sprague Dawley rat model for pharmacokinetic parameters. The goal is to determine responses in oral bioavailability to the Nano-PTX formulation and thus to determine whether the permeability enhancement predicted by our previous Caco-2 study will translate to improved oral bioavailability.

## 2. Materials and Methods

### 2.1. Materials

Paclitaxel (PTX; 33069-62-4; purity 99%) was obtained from LC Laboratories (Woburn, MA, USA). Ritonavir (internal reference) was purchased from LKT Laboratories Inc. (St. Paul, MN, USA). Taxol^®^ (an injectable solution of paclitaxel at 6 mg/mL) was purchased from BEDFORD Laboratories^TM^ (Bedford, MA, USA). Rubusoside (RUB; CAS 64849-39-4) was isolated from *Rubus suavissimus* S. Lee (*Rosaceae*) in our laboratory, and its structure was elucidated using nuclear magnetic resonance (NMR) and mass spectrometry analyses [3]; its purity was determined with HPLC to be greater than 98%. All HPLC-grade reagents, including methanol (Fisher Scientific, Pittsburgh, PA, USA), acetonitrile (Fisher Scientific, Pittsburgh, PA, USA), methyl tert-butyl ether (Acros Organics, Geel, Belgium), water (Fisher Scientific, Pittsburgh, PA, USA), and formic acid (Fisher Scientific, Pittsburgh, PA, USA), were purchased in the U.S. market. All other chemicals were reagent grade and used as received. Nylon 0.45 µM filters were purchased from Whatman (Maidstone, Kent ME14 2LE, UK). Agitation was obtained on a RAPIDVAP system (Labconco, Kansas City, MO, USA). Jugular vein pre-cannulated Sprague Dowley (SD) rats were obtained from Harlan Laboratory Co., Ltd. (Indianapolis, IN, USA).

### 2.2. Preparation of Water-Soluble Paclitaxel Formulation

We followed closely the procedure described in the in vitro studies [3] to prepare Nano-PTX. Briefly, appropriate amounts of RUB and PTX were weighed at a ratio of 50:1 weight/weight (*w*/*w*). Thereafter, 10 mL of absolute ethanol was added to the mixture, vortexed slightly, and heated in a water bath as needed to form a clear ethanol solution containing 2 mg/mL PTX and 100 mg/mL RUB. The ethanol solution was passed through 0.45 μm nylon filters (Whatman, Maidstone, Kent, UK) to eliminate insoluble impurities present in the solution. The ethanol solution was allowed to stand at room temperature (22 °C) for 60 min and was then evaporated to powder under reduced pressure at 50 °C and agitation in a RAPIDVAP system (Labconco, Kansas City, MI, USA). The powder of Nano-PTX was reconstituted with deionized water and passed through 0.45 μm nylon filters. The filtrate was diluted in mobile phase solvents and then injected into an HPLC system for analysis of PTX. The HPLC system and validated methodology were identical to those previously published by our group [3]. Briefly, the protocol ran an isocratic gradient of acetonitrile and water (52:48). A photodiode array detector recorded absorption in the range of 200–400 nm at 2 nm intervals. PTX was detected at 230 nm and RUB at 215 nm. Details can be found in the paper by Liu et al. [3].

Particle size measurement of the Nano-PTX formulation was performed on a Nanotrac NPA250 dynamic light scattering (DLS) apparatus (Microtrac, Inc., Montgomeryville and York, PA, USA) capable of measuring particle sizes ranging from 0.8 nm to 6.5 µm. The instrument software (FLEX 2011) reports particles detected based on volume, number, and area. We choose to report the volume of particles. A nanoparticle dispersion was prepared by weighing 300 mg of the Nano-PTX powder and then bringing it to 1 mL in deionized water. The ambient temperature was recorded as 19.0 °C. Each sample was run 3 times at a 90° scattering angle with durations of 100 s. Particle size per run was averaged over the particle size distribution. Particle size for each sample was averaged over the 3 runs.

### 2.3. Animal Selection and Housing

Jugular vein pre-cannulated male Sprague Dawley (SD) rats were chosen in this animal experiment. This surgically modified rodent model allows the collection of sufficient blood volume from the same rodent at multiple time points at minimal stress levels [11]. The surgical placement of round-tipped catheters was performed by the vendor (Harlan Laboratory, Indianapolis, IN, USA). Pre-cannulated male rats weighing between 250 and 300 g were housed individually in cages in a quarantine room. After 4-day acclimation, they were released to the assigned experiment room in the vivarium. To keep catheters unobstructed during the experimental period, the catheters were flushed with heparinized glycerol (500 U/mL) every 5 days. All the studies on animals were approved by the Louisiana State University Agricultural and Mechanical (LSU A&M) Institutional Animal Care and Use Committee under protocol number 14-044. The LSU A&M is accredited by AAALAC International.

### 2.4. Pharmacokinetic Study

The pharmacokinetic study was performed as described previously for flavonoid and ceramide using the same animal model [12]. A total of 36 pre-cannulated male SD rats were used. Rats were randomly divided into six groups with six rats per group. Three groups of rats were randomly assigned to receive an oral administration of Taxol^®^ solution at a dose of 20 mg/kg body weight, Nano-PTX aqueous solution at 20 mg/kg body weight, or deionized water (blank control). Oral administration was conducted via oral gavaging 1.0 mL of a formulation with the paclitaxel amount adjusted based on the body weight of each rat to conform to the 20 mg/kg dosing regimen. The other three groups of rats were randomly assigned to receive an intravenous injection through cannulation of Taxol^®^ solution at a dose of 2 mg/kg body weight, Nano-PTX aqueous solution at a dose of 2 mg/kg body weight, or physiological saline (blank control). The injection volume was all set to be 100 µL, and the paclitaxel amount was adjusted based on the body weight, so it delivered 2.0 mg/kg to each rat. According to the appropriate dose and the composition of paclitaxel formulations, all samples containing PTX for oral and intravenous administration were prepared. Before intravenous injections via cannulation tubes, all formulations passed through a 0.22 µm nylon filter to ensure exclusions of particles and microbial organisms. Fasting began 12 h before and ended 2 h after drug administration. Blood (250 µL) was collected via the cannulation tube at 0, 0.25, 0.5, 0.75, 1, 1.5, 2.5, 4, 6, 8, 12, and 24 h after administration and immediately centrifuged at 4000 rpm for 10 min to obtain plasma. The resulting plasma samples were stored at −18 °C until instrumental analyses. Maximum plasma concentration (C_max_) and the time to the maximum plasma concentration (T_max_) were determined directly, and other pharmacokinetic parameters, including elimination half-life (t_1/2_), were calculated by using the noncompartmental data analysis program Kinetica v5.0 (Thermo Fisher Scientific, Waltham, MA, USA). The area under the plasma concentration–time curve (AUC) from zero to infinity (AUC_0–24_) was calculated by means of the linear trapezoidal rule with extrapolation to infinity with terminal elimination rate constant (K_e_). Absolute oral bioavailability was calculated based on the ratio of AUCs between oral and intravenous administration, adjusted by the dosing regimen. The absolute bioavailability (F_abs_) was calculated using the following equation: F_abs_ = (AUC_PO_ × D_IV_)/(AUC_IV_ × D_PO_) × 100, where PO denotes oral route, IV denotes intravenous route, and D stands for dose. Relative bioavailability (F_rel_) compares two formulations administered in the same route and is calculated in the following equation: F_rel_ = (AUC_A_ × D_B_)/(AUC_B_ × D_A_) × 100, where the subscripts A and B indicate formulation A or B, and D denotes the dose of formulation B or A.

### 2.5. Instrumental Analysis of Paclitaxel

Prior to analysis, the frozen plasma was thawed at room temperature. The plasma from each rat was extracted and homogenized with 20 µL of 500 ng/mL ritonavir (International Standard, or I.S.) solution on a vortex mixer (Qilinbeier Vortex, Houston, TX, USA) for 2 min, then mixed with 1.2 mL methyl tert-butyl ether for 30 min and, finally, centrifuged at 12,000 rpm (18,000× *g*) for 10 min, resulting in organic and water layer separation. The organic layer was carefully transferred to a clean tube and evaporated to dryness under pressured nitrogen gas blowing at room temperature. The residue was reconstituted with 100 µL of methanol, centrifuged at 12,000 rpm (18,000× *g*) for 10 min, filtered with a 0.22 µm filter, and transferred to an injection vial. The instrumental analysis was performed using a Waters (Waters, Milford, MA, USA) HPLC-MS system equipped with a 1525 pump, 717 autosampler, 2996 photodiode array, and EMD1000 mass spectrometer with an electrospray source. The signal acquisition and peak integration were performed using Empower2 workstation software (v2.0). Chromatographic separations were achieved using a C_18_ column (2.0 × 150 mm, 5 µm Phenomenex, Torrance, CA, USA) at 30 °C controlled with a column chamber. The mobile phase consisted of HPLC-grade methanol and water containing 0.02% formic acid (*v*/*v*) and run in a linear gradient from 72% to 78% methanol in 0.02% formic acid water for 12 min and then kept at 78% methanol in 0.02% formic acid water for 15 min. The flow rate of the mobile phase was 0.26 mL/min, and the injection volume of the sample was 20 µL.

The mass spectrometer was operated in positive electrospray ionization (ESI+) mode. Nitrogen was used as the desolvation gas at a flow rate of 400 L/h and cone gas at a flow rate of 50 L/h. ESI+-MS was in positive mode and SIR scan at paclitaxel *m*/*z* = [M + Na] + 876.5 and ritonavir *m*/*z* = [M + Na] + 743.5. The desolvation temperature was 400 °C. The source temperature was 120 °C. The capillary voltage and the cone voltage were set at 3.5 kV and 35 V, respectively.

The standard stock solutions of paclitaxel and ritonavir were prepared in methanol at a concentration of 1000 ng/mL and 500 ng/mL, respectively. The working standards were prepared by diluting the paclitaxel stock solution in methanol to 1.25, 2.5, 5, 7.5, 10, 25, 50, 75, 100, and 250 ng/mL.

### 2.6. Statistical Analysis

Pharmacokinetic parameters of paclitaxel were obtained using the Kinetica software (v4.2; Innaphase, Philadelphia, PA, USA) using the non-compartmental analysis method. Statistical differences were determined by using Student’s *t*-test (SAS, Cary, NC, USA). Data were expressed as mean ± SE (standard error) unless otherwise specified. The significance of all tests was set at *p* ≤ 0.05.

## 3. Results

### 3.1. Water-Soluble Paclitaxel (Nano-PTX) Preparation

Following the previously used method [3], the Nano-PTX formulation was successfully prepared. The powder of the Nano-PTX formulation contained 1.96% PTX and 98.04% RUB by weight, respectively. When the Nano-PTX powder (306 mg) was redissolved in 0.89 mL of deionized water, PTX was 6.0 mg/mL (Table 1). This concentrate was diluted to 3 mg/mL, 1 mg/mL, 500 µg/mL, and 100 µg/mL in deionized water or saline. The dilutions were clear and transparent. HPLC analyses confirmed the concentrations of PTX at each dilution proportional to the dilution factor. The agreement between HPLC analytical results and initial feeding of the two ingredients at 50:1 *w*/*w* suggests full drug loading and complete micellar encapsulation. Taxol^®^ came in an injectable concentrate at 6.0 mg/mL. When diluted in saline to 1 mL, Taxol^®^ provided PTX at 6.0 mg coupled with 527 mg polyoxyethylated castor oil and 0.497 mL ethanol. Dilutions in deionized water and saline by a factor of 2, 6, 12, and 60 remained clear. PTX concentrations were not analyzed. The Nano-PTX water concentrate was clear and transparent and remained so after being diluted to 2 mg/mL PTX in deionized water containing 100 mg/mL RUB. Measured on the DLS particle size analyzer, this dispersion was found to contain nanoparticles ranging from 2.0 nm to 9.5 nm. Within this range, the smallest particle size was 2.0 nm at 6.5%, 2.5 nm at 31.5%, 4.0 nm at 26.5%, 5.0 nm at 18.75%, 6.0 nm at 11.5%, 7.0 nm at 3.75%, 8.0 nm at 1.25%, and the biggest was 9.5 nm at 0.25% (Figure 1a). The cumulative distribution reached 100% thereafter (Figure 1b), indicating the dispersion was monomodal with only one peak displaying a narrow cluster of similar-sized particles no more than 7.5 nm apart. The peak averages 4.7 ± 0.7 nm in diameter. These nanoparticles are referred to as spherical nano-micelles. The two formulations for this study differ in physical form. Nano-PTX was supplied as a powder for reconstitution in water for oral and intravenous administration. Taxol^®^ came in dilutable liquid for oral and intravenous administration (Table 2).

### 3.2. Paclitaxel Pharmacokinetics

Oral administration of two paclitaxel formulations at the same dose of 20 mg/kg paclitaxel resulted in significant differences in some parameters but not others between the Nano-PTX and Taxol^®^ for mulations. Comparing the two formulations, Nano-PTX showed significantly higher C_max_ (1.5-fold), faster T_max_ (1-fold), and overall, more (1.9-fold) plasma drug than Taxol^®^ (Table 3). Other parameters such as T_1/2_, MRT, CL, and V_dss_ were not significantly different between the two formulations. After oral administration, paclitaxel concentration from the Nano-PTX formulation increased rapidly and reached a maximum of 69.9 ± 6.47 ng/mL at 45 min, whereas that from the Taxol^®^ formulation reached the maximum of 46.2 ± 5.8 ng/mL at 90 min (Figure 2; Table 3). Over the 24 h following oral administration, paclitaxel concentrations delivered by the Nano-PTX formulation were higher than those delivered by the Taxol^®^ formulation except at a couple of time points where the concentrations overlapped. The AUC of PTX, representing the total paclitaxel amount in the circulating bloodstream, was 332.3 ng min/mL from the Nano-PTX formulation compared with the lower amount of 174.4 ng min/mL from the Taxol^®^ formulation. The apparent plasma half-life of paclitaxel delivered by the Nano-PTX formulation was 6.5 h, whereas the half-life was 3.1 h when it was delivered by the Taxol^®^ formulation at an equivalent dose. The mean residence time (MRT) of paclitaxel was 8.6 h in the Nano-PTX formulation compared to the shorter 4.9 h in the Taxol^®^ formulation. Nano-PTX cleared the bloodstream slower at a rate of 72.2 ± 10.4 mg/L within 24 h compared to the faster 129.6 ± 19.7 mL/h exhibited by the Taxol^®^ formulation. The apparent volume of distribution at a steady state was very similar and showed insignificant differences between the two formulations.

Following intravenous injection of paclitaxel in the form of Nano-PTX or Taxol^®^, paclitaxel immediately reached its maximum concentration, regardless of formulation. Thereafter, concentrations decreased rapidly within 90 min and slowed afterward. The two intravenous formulations displayed nearly identical patterns during the 24 h of the study period (Figure 3). Other than the slightly higher (not significant) paclitaxel concentrations at nearly all time points (higher AUCs displayed by the Nano-PTX formulation), the pharmacokinetic parameters were very similar to the Taxol^®^ formulation (Table 4). Even though paclitaxel from the Nano-PTX formulation reached its maximum concentration at a significantly shorter time, the calculated time gap was only 3 min apart (0.25 vs. 0.3 h) from the Taxol^®^ formulation. One noticeable difference was the clearance rate, which showed the Nano-PTX formulation was lower than the Taxol^®^ formulation (1.2 vs. 2.1 mL/h/kg). The same trend was observed in the oral administration. Still, there were differences despite insignificance. The maximum plasma concentration (C_max_) of Taxol^®^ was 492.3 ± 118.6 ng/mL, lower than 516.3 ± 175.2 ng/mL of Nano-PTX. The AUC of nano-PTX was 1608.4 ng min/mL compared with 1036.8 ng min/mL of Taxol^®^. The apparent plasma half-life was closest, 11.8 h vs. 11.3 h, between the Nano-PTX and Taxol^®^ formulations.

### 3.3. Oral Bioavailability

Absolute oral bioavailability of paclitaxel delivered with the Nano-PTX and Taxol^®^ formulations was 1.7% and 1.3%, respectively (Table 5). Comparing the two formulations, the Nano-PTX formulation delivered 88% more paclitaxel into the bloodstream than the Taxol^®^ formulation.

## 4. Discussion

Delivering a BCS IV compound such as Paclitaxel is challenging, especially in an oral route. Paclitaxel demonstrates extremely poor solubility and low permeability. The Taxol^®^ formulation resolved the poor solubility issue by using Cremophor EL (polyoxyethylated castor oil derivatives) and dehydrated alcohol, enabling intravenous administration to treat cancers. This formulation employs the techniques of co-solvency and emulsion that reduce surface tensions. Other co-solvents and surfactants include dimethylacetamide, propylene glycol, low molecular weight polyethylene glycols, Tween 80, and solutol HS 15. Using complexing excipients such as hydroxypropyl-β-cyclodextrin and captisol [13] is also a common tool for achieving adequate solubility.

Paclitaxel has very low intrinsic solubility in water at a reported 0.1 µg/mL [14], 520-fold below the minimum required solubility of 52 µg/mL for successful drug development [15]. Although exploiting the polymorphs increased paclitaxel’s solubility 60-fold to 6.05 µg/mL, the level of solubility is still inadequate. Other methods increased the solubility of paclitaxel. Strategies include using surfactants such as d-tocopherol polyethylene glycol 1000 succinate (TPGS) 400 [16], reducing the particle size by enclosing paclitaxel into microemulsions or nanoemulsions [17], inserting paclitaxel into the cavity of cyclodextrin molecules [18,19], or creating nanoparticles containing paclitaxel molecules [20]. Moreover, co-solvency, emulsification, micellization, liposome formation, and non-liposomal lipid nanocarriers (solid lipid, microspheres, nano-capsules, nanoemulsion) were explored with some success [21]. The first nanocarrier drug, Doxil^®^, approved by the FDA, is a great example of such success. The well-known chemotherapeutic drug Taxol^®^ was formulated with the joint use of an emulsifier and a co-solvent, i.e., a blend of Cremophor EL (polyoxyl 35 castor oil), and dehydrated alcohol at a 1:1 volume ratio. Despite progress in addressing a determinant factor of solubility, other issues brought side effects such as low drug loading and potential toxicity of these excipients to achieve undesirable levels of solubility.

The natural product rubusoside has never been used as a solubilizer for dissolving active pharmaceutical ingredients. It is a steviol glycoside isolated from the plants of *Rubus suavissimus* and *Stevia rebaudiana*. As its structure displays, the lipophilic steviol core and hydrophilic glucose side chain potentiate a surfactant property. In an aqueous solution, these surfactant molecules tend to self-assemble and form micelles [22]. The detection of approximately 5 nm-diameter spheres confirmed the formation of micelles in this study. It is plausible that the lipophilic paclitaxel is embedded inside the RUB-formed micelles being attracted to the lipophilic steviol core in an unknown configuration. It has long been known that the low oral bioavailability of paclitaxel is related to poor solubility and high affinity of P-gp in the GIT. Once recognized by P-pg, free paclitaxel molecules are easily transported out of the enterocytes back to the gut lumen. However, paclitaxel may not be in a free form as long as it is embedded inside the RUB-assembled micelle. It is reasonable to speculate that paclitaxel may have been shielded from being recognized as its free structure was no longer exposed to P-pg. If the micelle containing paclitaxel remains intact, it is possible that paclitaxel may evade the P-gp. If paclitaxel comes out of the micelles, then the free molecule is likely captured by P-gp and removed. This might be the case with the Taxol^®^ formulation (an emulsion) where paclitaxel is dispersed in water probably with little or no disguise. How effective rubusoside is in shielding paclitaxel from the P-pg is unclear. It would have to be much more apparent in the bioavailability results to suggest this as a viable explanation. The in vitro Caco-2 result undoubtedly supported the permeability enhancement of the Nano-PTX formulation over the Taxol^®^ formulation [3], but it predicts nothing past the initial absorption. In vivo, other factors come into play, especially first-pass metabolism. Based on the result of this study, it appears that paclitaxel-loaded micelles formed with RUB may not have survived the first-pass metabolism. The paclitaxel may have been released/dissociated from the micelles thus losing the protection of the micellar structure. The Nano-PTX formulation did show higher oral bioavailability than the Taxol formulation, but the improvement was not massive. Had liver-related enzyme assays been conducted, they could have shed some light on where the improvement with the Nano-PTX formulation ended.

Both formulations of Nano-PTX and Taxol^®^ achieved solubilization of 6 mg/mL in water solution. However, as described above, the solubilization mechanism and effect differ. The Nano-PTX formulation enclosed paclitaxel in micelles that interact directly with water molecules through the polar glycosides. The Taxol^®^ formulation disperses paclitaxel in water via molecular interaction between water-miscible alcohol and surfactant. These different features, i.e., free or enclosed molecules may have been translated to differences in some pharmacokinetic parameters. The concentration–time curves (Figure 2 and Figure 3) clearly show a consistent and persistent outperformance for the Nano-PTX formulation over the Taxol^®^ formulation across routes of administration (i.e., oral and intravenous). Although the mechanism is unknown, the paclitaxel in nano-size, the micellar structure, and RUB may play a different role other than solubilization, which may have contributed to the observed improvement of oral bioavailability. At the same oral dose of 20 mg/kg and the same level of solubility (6 mg/mL), paclitaxel from the Nano-PTX formulation showed significantly higher C_max_, shorter T_max_, longer MR, and, overall, more (1.9-fold) plasma concentration than that from the Taxol^®^ formulation. Reasonable explanations for these intriguing differences are unavailable. One thing is certain; the use of RUB resulted in a more desirable pharmacokinetic profile. The pharmacokinetic parameters of paclitaxel intravenously administered were very similar between the two formulations and shared nearly identical patterns. Still, Nano-PTX delivered more paclitaxel (1.2-fold) into the circulating bloodstream over a 24-hour period than Taxol^®^.

Injecting water-soluble paclitaxel via the Nano-PTX formulation was not the objective of this study but rather a way to estimate absolute oral bioavailability. It was interesting to learn that the micellar structures can be injected. Injecting paclitaxel-loaded micelles presumably preserved the intact PTX-RUB micellar structure in the bloodstream, but how long they stayed and when the two broke apart remains a mystery. Many studies have proved that the intravenous colloid drug system can significantly change the pharmacokinetic properties of drugs. Our studies also indicated that the pharmacokinetic properties were markedly different between the two formulations. Paclitaxel delivered via the Nano-PTX formulation (micelles) exhibited a markedly delayed blood clearance compared with that via the Taxol^®^ formulation (emulsion). Micelles were found capable of significantly delaying clearance over the emulsion formulation [23]. Whether nanoencapsulation produced a controlled-release effect could be of interest to explore in an in vitro release assay. The initial burst effect right after intravenous injection observed for both formulations could cause an eventual toxic effect. However, given the similar pattern between the two formulations, this may not have happened. Similarly, the initial burst of low plasma drug level (about 1/9th) after oral administration during the first 2 h compared with the intravenous route is unlikely to cause eventual toxic effects. Regardless of the positive effect, injecting RUB-based micelles for drug delivery is not practical as RUB is not an FDA-approved drug excipient. However, its GRAS status promises an oral delivery.

It was assumed that paclitaxel within micelles remained in the micelles through the GIT track onto the epithelial surface. We don’t know if crossing the epithelial membrane was via micelles or free compounds. Regardless of the difficulties of overcoming poor solubility and low permeability, efforts to develop oral formulations for paclitaxel continue. This is because paclitaxel remains thus far one of the most potent chemotherapeutic compounds. The continued use of paclitaxel for inhibiting angiogenesis led to the FDA approval of anti-clogging stent operations (FDA: Paclitaxel-Coated Balloons and Stents for Peripheral Arterial Disease), demonstrating the sustained interest in old drugs for new uses [24]. Although the Nano-PTX formulation increased the permeability of paclitaxel by more than 3-fold over the Taxol^®^ formulation in the Caco-2 cell model, which justified this animal study, the oral bioavailability improvement fell short to less than 1-fold. Clearly, there were other factors involved in the animal study that were absent in the cell model. Missing from the cell model is the liver where cytochrome P450 enzyme family (CYP) CYPs are enzymes that account for up to 70% of drug metabolism [25], while P-gp is an efflux pump that extrudes drug substrates out of cells, and the inhibition of P-gp has been shown to increase blood level paclitaxel in a human clinical study [26], serving as proof that inhibiting P-gp is a plausible approach. It is reasonable to speculate that the enhanced permeability observed in the Caco-2 cell model of this Nano-PTX formulation was a result of evading P-pg through unknown mechanisms, e.g., disguising paclitaxel inside the micelles and/or inhibiting with RUB. CYP is not involved in the Caco-2 cell model but was in the animal model used in this study. It is assumed that the Nano-PTX micelles successfully countered the efflux pump effects and entered the liver via the hepatic portal vein where they underwent the well-known first-pass metabolism. There might be two fates. If paclitaxel was separated from its micellar structure and broke free, it could be degraded by CYP. Approximately 90% of paclitaxel is primarily metabolized in the liver by the cytochrome (CYP) P450 enzyme system, particularly involving the CYP2C8 and CYP3A4 isoenzymes [27,28]. If it remained in the micelles, it might be treated as an exogenous compound and sent to the biliary system for excretion [29]. Pre-systemic elimination in the intestinal wall and liver by cytochrome isoenzymes 3A4 and 2C8 was reported to cause the low oral bioavailability of paclitaxel. These possible pre-clearance actions reduced the amount of paclitaxel entering the circulating bloodstream, leading to lowered bioavailability. As a future follow-up study to investigate some mechanisms, the Nano-PTX formulation may be tested for its interaction with CYP to determine if including a CYP inhibitor in an upgraded formulation could mend the 2-fold gap between our previous Caco-2 permeability enhancement (3-fold) and oral bioavailability improvement of this study (less than 1-fold).

RUB is GRAS, and paclitaxel is an approved drug compound. Theoretically, both ingredients in this Nano-PTX formulation would deliver a potent drug compound orally in a safer and more convenient route of administration than injection. RUB is approved for natural food sweetener uses (US FDA GRAS Notices), and dietary consumption presented no adverse effects in rats up to 2 g/kg. Unfortunately, the oral bioavailability was found to be only 1.7%, a slight increase from the Taxol^®^ formulation. Normally, 1 to 2% oral bioavailability is regarded as low to warrant the development of this Nano-PTX formulation. At the current formulation, Nano-PTX is not promising for entering an oral medication pathway. As a food sweetener and GRAS, the feasibility of tuning the Nano-PTX formulation for a sublingual delivery would be interesting. Paclitaxel is among the most potent drugs, achieving IC_50_ at a tens of nanomolar concentration [30] compared to the tens of micromolar concentration of curcumin [31]. This potency advantage of paclitaxel may translate to a much smaller dosage for easy delivery. Sublingual delivery is an interesting option as it would deliver the drug without first-pass metabolism.

## 5. Conclusions

Overcoming the poor solubility and low permeability of paclitaxel by the novel creation of the Nano-PTX formulation in a previous cellular model prompted this animal study. We predicted oral bioavailability would be improved 3-fold, yet bioavailability was increased by only 88% over the Taxol^®^. We speculate that hepatic first-pass metabolism lowers systemic drug levels. Upgrading the current Nano-PTX formulation by including a CYP inhibitor could lower first-pass losses. Future studies with in vitro release assays to examine controlled release properties could also reveal important functionalities. Rubusoside as a solubilizer and permeation enhancer offered a promising novel alternative approach to current pharmaceutical excipients such as Cremophor EL, yet more work is needed. Applying RUB for other Class IV compounds and future studies is warranted.

## Figures and Tables

**Figure 1 pharmaceutics-16-01104-f001:**
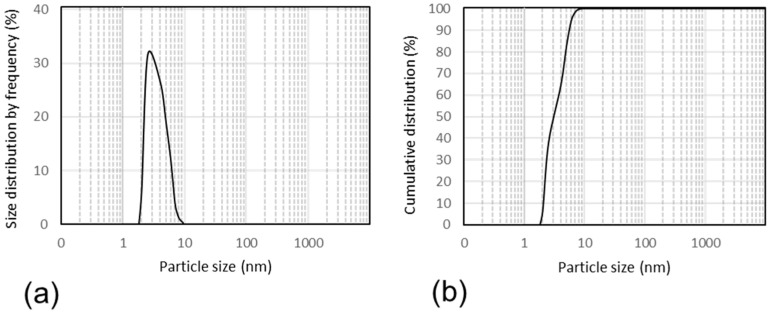
Particle size (nm) distribution detected in a 10% *w*/*v* RUB-solubilized paclitaxel (Nano-PTX) nanoparticle dispersion. (**a**): Size distribution by frequency. (**b**): Cumulative distribution.

**Figure 2 pharmaceutics-16-01104-f002:**
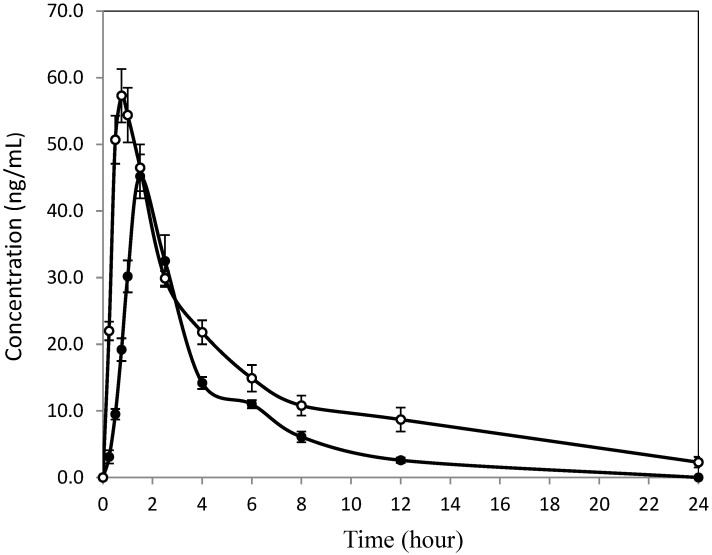
Time–concentration curves of Taxol^®^ (solid circle) and Nano-PTX (open circle) orally dosed at 20 mg/kg to Sprague Dawley rats. Taxol^®^ was diluted from the liquid drug with water and used as the PTX control. Nano-PTX formulation was prepared by reconstituting water-soluble powder with water and then diluted with water at the time of oral gavage. Each data point represents the mean of six replicates (n = 6). Vertical bars across each data point represent one standard error.

**Figure 3 pharmaceutics-16-01104-f003:**
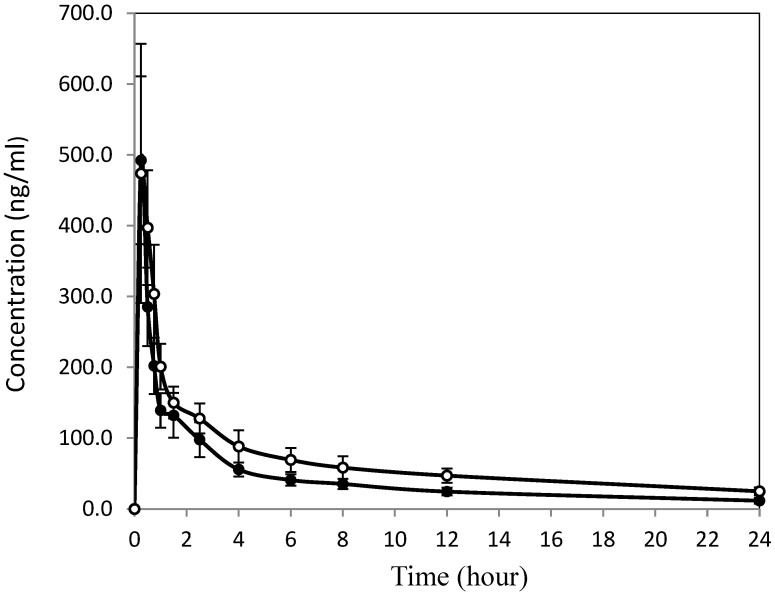
Time–concentration curves of two paclitaxel formulations intravenously administered at 2 mg/kg dose to Sprague Dawley rats. Taxol^®^ (solid circles) was diluted from the liquid drug and used as the control. Nano-PTX formulation (open circles) was prepared by reconstituting water-soluble powder with water and then diluted with saline at the time of injection via the pre-installed cannulation. Each data point represents the mean of six replicates. Vertical bars across each data point represent one standard error.

**Table 1 pharmaceutics-16-01104-t001:** Composition of paclitaxel formulations for oral and intravenous administration.

Component	Formulation	
	Taxol^®^	Nano-PTX
Paclitaxel ^1^ (mg)	6.0	6.0
Soluble paclitaxel ^2^ (mg)	6.0	6.0
PCO ^3^ (mg)	527.0	0
RUB (mg)	0	300
Absolute ethanol (mL)	0.497	0
Water (mL)	0	0.89
Total volume (mL)	1.0	1.0
Supply	5 mL/bottle	306 mg/bottle
Form	Liquid	Powder

^1^ The amount of paclitaxel in the formulation. ^2^ The concentration of paclitaxel that remained in the supernatant of a water solution as measured with HPLC. ^3^ PCO stands for polyoxyethylated castor oil.

**Table 2 pharmaceutics-16-01104-t002:** Paclitaxel formulations for oral and intravenous administration.

Formulation	Form	Supply	Add Water (mL)	Total Stock Volume	Solvent	Paclitaxel Concentration (mg/mL)
Taxol^®^	Liquid	5 mL/bottle	0	5.0 mL	Cremophor/Ethanol	6.0
Nano-PTX	Powder	1530 mg/bottle	4.45	5.0 mL	Water	6.0

**Table 3 pharmaceutics-16-01104-t003:** Pharmacokinetic parameters (mean ± standard error) for Sprague Dawley rats orally administered Taxol^®^ (positive control) or Nano-PTX (n = 6) at a dose of 20 mg/kg. Different letters following each mean of plasma samples for the same parameter indicate significant differences at *p* < 0.05, based on paired LSD *t*-test.

Parameter	Taxol^®^	Nano-PTX
AUC_0–∞_ (ng min mL^−1^)	172.4 ± 24.99 b	332.3 ± 81.32 a
AUC_0–24_ (ng min mL^−1^)	157.9 ± 20.82 b	283.1 ± 59.17 a
T_1/2_ (h)	3.1 ± 0.69 b	6.5 ± 1.88 a
MRT_0–∞_ (h)	4.9 ± 0.63 ab	8.6 ± 2.36 a
CL (mL/h kg^−1^)	129.6 ± 19.70 a	72.2 ± 10.35 a
Vdss (mL kg^−1^)	609.7 ± 80.67 b	539.2 ± 141.61 b
C_max_ (ng mL^−1^)	46.2 ± 5.82 a	69.9 ± 6.47 a
T_max_ (h)	1.7 ± 0.17 a	0.8 ± 0.15 b

**Table 4 pharmaceutics-16-01104-t004:** Pharmacokinetic parameters (mean ± standard error) for Sprague Dawley rats intravenously administered Taxol^®^ (control) or Nano-PTX (n = 6) at a dose of 2 mg/kg. Different letters following each mean of plasma samples for the same parameter indicate significant differences at *p* < 0.05, based on paired LSD *t*-test.

Parameter	Taxol^®^	Nano-PTX
AUC_0–∞_ (ng min mL^−1^)	1321.4 ± 337.33 b	1985.4 ± 438.14 ab
AUC_0–24_ (ng min mL^−1^)	1036.8 ± 214.52 b	1608.4 ± 352.22 ab
T_1/2_ (h)	11.8 ± 3.26 a	11.3 ± 0.93 a
MRT_0–∞_ (h)	13.5 ± 3.69 b	13.8 ± 1.04 b
CL (mL/h kg^−1^)	2.1 ± 0.39 a	1.2 ± 0.23 b
Vdss (mL kg^−1^)	25.67 ± 8.25 a	17.21 ± 3.59 a
C_max_ (ng mL^−1^)	492.3 ± 118.56 a	516.3 ± 175.16 a
T_max_ (h)	0.25 ± 0 c	0.3 ± 0.05 b

**Table 5 pharmaceutics-16-01104-t005:** Absolute oral bioavailability (%) of Nano-PTX and Taxol^®^ in Sprague Dawley rats.

Formulation	Absolute Oral Bioavailability (%)
Nano-PTX	1.7
Taxol	1.3

## Data Availability

The datasets presented in this article are not readily available due to technical/time limitations or privacy restrictions. Requests to access the datasets should be directed to the corresponding author.

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
