# Peer review of "Solubilization of Paclitaxel with Natural Compound Rubusoside toward Improving Oral Bioavailability in a Rodent Model"

_pharmaceutics, 2024, doi:10.3390/pharmaceutics16081104_

Round 1

Reviewer 1 Report

Comments and Suggestions for Authors

The present study has a promising hypothesis but failed to demonstrate it. During my revision, I identified numerous aspects that must be addressed to improve the document’s quality, which follows:

The natural molecule must be identified in the title. Furthermore, the current version does not represent the concrete findings of this study and must be enhanced.

In the abstract, essential information is missing, such as the composition of nano-micelles; concentration of rubusoside, method of micelles preparation, concentration of paclitaxel; doses used in the animals.

Keywords should be replaced with terms that were not used in the title or abstract; the authors could explore the Mesh terms available on the Pubmed library;

The introduction section is interesting but too long and should be reduced; the objective presented in the last paragraph differs from the one in the abstract. Please revise it and clarify such aspects. Based on the information provided in the introduction, the authors must reinforce the novelties of this study in the abstract.  The present form induces the reader to the idea that in vitro tests were performed.

What are the final concentrations of PTX and RUB in the nano-based formulation? It must be presented in the text;

What is the HPLC methodology to detect PTX and RUB? Was it standardized or validated?  Do the authors know about the PTX stability under the conditions of formulation preparation? It is essential to verify the final molecule concentration after preparing the formulation.

The number of ethical protocol approvals must be provided;

To be suitable for intravenous route formulations, they must be sterile. Please better describe and discuss how it was achieved;

What was the PXT dose administered to the animals?

The tables do not follow the Pharmaceutics’ guidelines for tables.

The stability in SGF was not described in the methods section;

The caption of Figure 3 must be enhanced; what is the meaning of the letter? The significant statistical differences should be better stated; the PTX concentration is in plasma samples;

In line 326, the authors stated that the first-pass metabolism could negatively affect the formulation, but the intravenous administration presents a similar PK profile. It must be better discussed;

The discussion is interesting and highlights this study's potential limitations, which is one of its major strengths. However, I must recognize that the possible slight effect of nanoencapsulation may be due to the system's controlled-release properties. To better investigate, the authors must conduct in vitro release assays using biomimetic mediums (SGF and simulated plasma fluid). Exploring such evidence will make the study more applicable.

The conclusion section is too long and must be reduced;

Comments on the Quality of English Language

There are numerous typographical errors in the manuscript. Abbreviations are used inconsistently and some are not used properly, particularly in the discussion section. Additionally, some paragraphs contain non-technical language. Please revise these aspects to improve the quality of the manuscript.

Reviewer 2 Report

Comments and Suggestions for Authors

The study focuses on the evaluation of the pharmacokinetics in model rats of a a micellar formulation of paclitaxel using rubososide (RUB).

The authors should provide additional evidences that micelles loading paclitaxel can be formed.

At least, the particle size of RUB dispersion in water should be measured, and to demonstrate that RUB can  decrease the surface tension.

It is not clear which is the final loading of the paclitaxel inside micelles since during the processing the preparation is filtered two times before injecting into HPLC (line 127) and before performing particle size (line 133). The authors should report the encapsulation efficiency and drug loading.

Line 131-133 Why 510 mg of Nano-PTX in 5 mL of water should contain 2 mg of PTX and 100 mg of RUB? How it was calculated?

Line 219-220 the sentence should be supported by experimental data or by a reference reporting the experimental data.

Line 221 Why Nano-PTX was coupled with 300 mg of RUB? Is 6 mg the dose for paclitaxel of the weight of Nano-PTX? In the case, it would be the dose (Table 2), how 6 mg was determined?

Is the particle size distribution in Figure 2 in volume (%) or number (%)?

Author Response

Plese see the attachment.

Reviewer 3 Report

Comments and Suggestions for Authors

Authors proposed a paper entitled “Solubilization of paclitaxel by a natural compound towards im-proving oral bioavailability in a rodent model” for the publication in Pharmaceutics, MDPI.

The paper requires major revisions and the addition of explanations, where requested.

Here is the list:

Line 16. “oral delivery”. Let us talk about also “bioavailability after oral administration”.

Line 21. “Taxol after oral and intravenous” please remove double space here.

Line 24. “

 mean particle size was 4.67 ± 0.07 nm” please check the value of the standard deviation. Moreover, we have two decimal digits of nanometers, that surely come from averages calculated by the instrument. I would approximate to unity or at least to one single decimal digit.

Line 34. Paclitaxel is a well-known molecule in this pharmaceutical field; therefore, I find it not so useful to represent the molecule here.

Line 35. “Taxus brevifolia” shouldn’t this be reported in italics?

Line 64. Could you please provide more references and information about “lipid carriers (microspheres, nano-capsules)”?

Line 70. Figure 1b could be also avoided, as well. It does not add significant information for the aims of this paper.

Figure 1 caption is a figure; however, it should be a written text in the manuscript. This comment is valid for all the images reported in the manuscript.

CAS numbers of materials used for formulations preparations should be reported.

Line 121. “mg/ml PTX and 100 mg/mL” please uniform to “mL”

Line 131. “0.8 nm to 6.5 micron” please use “µm”.

Line 135. “Particle size was averaged”. Please be more specific: was all the PSD curve averaged or the D[50] only?

Line 159. “two hours” please use numbers.

Line 165. “program, Kinetica” the previous “comma” is not necessary.

The caption of Table 1 should be written in the manuscript, not just be part of the figure. The quality is low. This is also valid for subscripts of the table. The table itself should be formatted as a table, not as a figure.

Line 179. “20 μl of 500 ng/ml” uniform “ml” to “mL”, as indicated before.

Line 215. “Following the previously used method [2] with slight modifications” are these modifications reported in materials and methods section of the present paper?

Figure 2. It would be better to extract raw data and draw particle size distribution using one of the common software such as Sigma Plot or Origin or more others. It would help correctly detecting the shape and main characteristics of this PSD. Moreover, the name of the y-axis should be “Particles, %” or even “Frequency, %”.

Table 2 and Table 3 should be reported in Table format, not Figures, since the quality is low. Please, also follow the guidelines of this journal.

Calibration curves information could be reported in Supplementary materials, they are not useful here.

Table 4 should be transformed using table format. Moreover, the symbols “a” and “b” should be defined in the subscripts of this table.

Figure 3 and Figure 4. How to authors explain the eventual toxic effects of the initial burst registered at the first 2 hours from administration? Authors never talk about “burst” effects in this paper. Can you add some comments on these effects?

Table 6 and Table 7 could be merged after transforming them into a word table (we cannot accept them as figures).

Comments on the Quality of English Language

Quality of English is quite good.

Round 2

Reviewer 1 Report

Comments and Suggestions for Authors

The document was consistently revised. The current form is adequate for acceptance.

Author Response

Response: We truly appreciate your constructive comments and suggestions!

Reviewer 2 Report

Comments and Suggestions for Authors

1) Line 233-235 “Measured on the DLS particle size analyzer, this water solution was found to contain water-soluble nanoparticles ranging from 1 to 10 nm (Figure 1)”. This sentence is not correct. It is not a solution but it is a nanoparticles dispersion.

2) If “The ethanol solution was passed through nylon 0.45 μm nylon filters (Whatman, Maidstone, Kent, UK) to eliminate larger particles present in the solution” (line 111-112), how can be that a “full drug loading and complete micellar encapsulation” (line 225) occurred?

Author Response

1) Line 233-235 “Measured on the DLS particle size analyzer, this water solution was found to contain water-soluble nanoparticles ranging from 1 to 10 nm (Figure 1)”. This sentence is not correct. It is not a solution but it is a nanoparticles dispersion.

Response: Technically, you are right. It is a dispersion of nanoparticles, not a molecular interaction between water and paclitaxel. It has been corrected on Line 232.

2) If “The ethanol solution was passed through nylon 0.45 μm nylon filters (Whatman, Maidstone, Kent, UK) to eliminate larger particles present in the solution” (line 111-112), how can be that a “full drug loading and complete micellar encapsulation” (line 225) occurred?

Response: It’s a good point to clarify. At the ethanol stage, we assume all compounds in the formulation are dissolved in the solvent (ethanol). Therefore, filtration will not block the passage of any compounds but the impurity that does not dissolve. We added so it reads “to eliminate insoluble impurities” (Line 99) to clarify the purpose of this filtration. Moreover, it is the nanomicelles that were formed in water after the removal of ethanol that constitute the full drug loading and complete micellar encapsulation. Any paclitaxel that falls outside of the micelles in water would have been eliminated from the 0.45 um filters before HPLC injection. HPLC result supports the full loading and complete micellar encapsulation as the feeding concentration agrees with the analysis.   

Reviewer 3 Report

Comments and Suggestions for Authors

Authors responded to my issues point by point. Suggestions and advice were followed and clearly commented or modified, where needed. I would only modify the diagram in Figure Nr. 1; authors could adjust the scales of figures, or separate cumulative diagram by frequency diagram, using raw data. More simply, authors could just leave frequency distribution diagram.

Author Response

Authors responded to my issues point by point. Suggestions and advice were followed and clearly commented or modified, where needed. I would only modify the diagram in Figure Nr. 1; authors could adjust the scales of figures, or separate cumulative diagram by frequency diagram, using raw data. More simply, authors could just leave frequency distribution diagram.

Response: Thank you. We separated the two in a modified Figure 1 and made corresponding changes in the caption Line 215-217.  It’s a better idea. Thank you!